



Atmospheric
Measurement
Techniques

# Evaluation of satellite retrievals of liquid clouds from the GOES-13 imager and MODIS over the midlatitude North Atlantic during the NAAMES campaign

David Painemal[1,2], Douglas Spangenberg[1,2], William L. Smith Jr.[2], Patrick Minnis[1,2], Brian Cairns[3], Richard H. Moore[2], Ewan Crosbie[1,2], Claire Robinson[1,2], Kenneth L. Thornhill[1,2], Edward L. Winstead[1,2], and Luke Ziemba[2]

[1]Science Systems and Applications, Inc., Hampton, VA, US
[2]NASA Langley Research Center, Hampton, VA, US
[3]NASA Goddard Institute for Space Studies, New York, NY, US

**Correspondence:** David Painemal (david.painemal@nasa.gov)

**Abstract.** Satellite retrievals of cloud droplet effective radius ($r_e$) and optical depth ($\tau$) from the Thirteenth Geostationary Operational Environmental Satellite (GOES-13) and the Moderate Resolution Imaging Spectroradiometer (MODIS) aboard Aqua and Terra, based on the Clouds and the Earth's Radiant Energy System (CERES) project algorithms, are evaluated with airborne data collected over the midlatitude boundary layer during the North Atlantic Aerosols and Marine Ecosystems Study (NAAMES). The airborne dataset comprises in situ $r_e$ from the Cloud Droplet Probe (CDP) and remotely sensed $r_e$ and $\tau$ from the airborne Research Scanning Polarimeter (RSP). GOES-13 and MODIS (Aqua and Terra) $r_e$ values are systematically greater than those from the CDP and RSP by at least 4.8 (GOES-13) and 1.7 μm (MODIS) despite relatively high linear correlation coefficients ($r = 0.52$–$0.68$). In contrast, the satellite $\tau$ underestimates its RSP counterpart by $-3.0$, with $r = 0.76$–$0.77$. Overall, MODIS yields better agreement with airborne data than GOES-13, with biases consistent with those reported for subtropical stratocumulus clouds. While the negative bias in satellite $\tau$ is mostly due to the retrievals having been collected in highly heterogeneous cloud scenes, the causes for the positive bias in satellite $r_e$, especially for GOES-13, are more complex. Although the high viewing zenith angle ($\sim 65°$) and coarser pixel resolution for GOES-13 could explain a $r_e$ bias of at least 0.7 μm, the higher GOES-13 $r_e$ bias relative to that from MODIS is likely rooted in other factors. In this regard, a near-monotonic increase was also observed in GOES-13 $r_e$ up to 1.0 μm with the satellite scattering angle ($\Theta$) over the angular range 116–165°; that is, $r_e$ increases toward the backscattering direction. Understanding the variations of $r_e$ with $\Theta$ will require the combined use of theoretical computations along with intercomparisons of satellite retrievals derived from sensors with dissimilar viewing geometry.

## 1 Introduction

Cloud properties estimated from satellite passive sensors have been crucial in advancing our knowledge of the role of clouds in the climate system and the Earth's energy budget (e.g., Loeb et al., 2009; Kato et al., 2011). The unprecedented global view from space has been facilitated by a constellation of more than a dozen satellites equipped with visible and infrared imagers suitable for the derivation of cloud properties. Among the various satellite sensors orbiting Earth, the Moderate Resolution Imaging Spectroradiometer (MODIS) on Terra and Aqua is the most widely used in cloud and climate research due to its high radiometric performance and relatively high pixel resolution, as well as the ability to provide nearly global spatial coverage by combining the multiple daily satellite overpasses. Complementary to MODIS, a number of geostationary satellites with adequate sensor wavelengths for deriving cloud properties comparable to MODIS are currently applied by various remote sensing

groups around the world to detect clouds and derive cloud phase, effective radius, optical depth, liquid/ice water path, and height (Stubenrauch et al., 2013; Roebeling et al., 2015). These geostationary cloud properties are receiving increased attention as their high temporal resolution allows for continuous monitoring of cloud systems, making the datasets ideal for numerous weather applications, including nowcasting and data assimilation (e.g., Benjamin et al., 2016; Jones et al., 2018).

Passive-based cloud algorithms typically rely on a visible channel for retrieving cloud optical depth (the vertically integrated cloud extinction coefficient) and an absorbing near-infrared channel for estimating cloud effective radius ($r_e$, the ratio of the third to the second moment of the droplet size distribution), which, in turn, can be utilized for indirectly estimating liquid water path. Numerous studies have documented factors that can possibly bias the passive satellite cloud retrievals based on bi-spectral algorithms, including among others subpixel variability, clear-sky contamination, solar and viewing angles effects, and three-dimensional radiative effects (e.g., Marshak et al., 2006; Kato et al., 2006; Zhang et al., 2012). Despite these sources of uncertainty, comparisons between in situ aircraft data and MODIS retrievals for marine stratocumulus clouds have shown excellent correlations for effective radius, optical depth, and liquid water path in the eastern Pacific and northeast Atlantic (Painemal and Zuidema, 2011; Painemal et al., 2012; Noble and Hudson, 2015; Zhang et al., 2017). In contrast, Ahn et al. (2018) found poor agreement between MODIS cloud effective radius and airborne cloud probe measurements over the Southern Ocean in winter. Unfortunately, the limited number and complexity of the samples in Ahn et al. (2018) prevented further inferences, an issue that illustrates the challenges of evaluating satellite observations in middle and high latitudes. Recently, Kang et al. (2021) evaluated MODIS cloud retrievals over the Southern Ocean in summer for overcast scenes, finding a relatively good agreement comparable to other assessments over the subtropics.

In situ and remotely sensed aircraft observations of cloud properties are key for evaluating cloud retrievals; however, sparse sampling and observational uncertainties hamper the satellite bias quantification. Optimal airborne measurements for assessing satellite observations should incorporate data redundancy, samples taken at different levels within the cloud, and use of observations within minutes of the satellite overpass time. Data redundancy helps minimize the misinterpretation of biases in satellite observations, whereas cloud vertical sampling allows for a more adequate comparison with satellite products, especially retrieved particle size, which is primarily contributed by a few optical depths from the cloud top (Platnick, 2000). Here we take advantage of aircraft measurements taken over the midlatitude North Atlantic during the North Atlantic Aerosols and Marine Ecosystems Study (NAAMES, Behrenfeld et al., 2019), which employed a sampling strategy well suited for evaluating satellite ob-

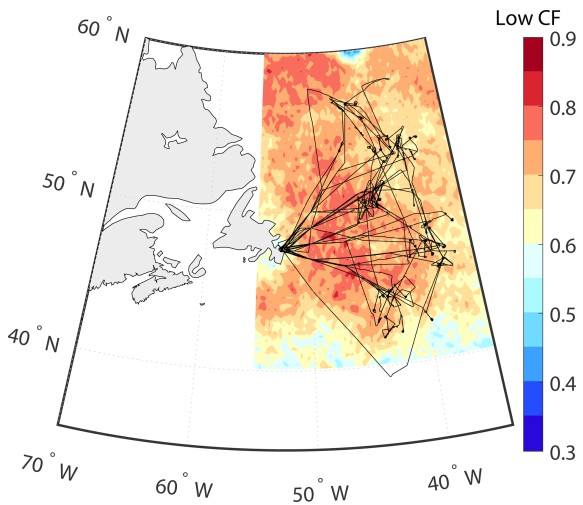

**Figure 1.** Mean Aqua MODIS low-cloud cover and aircraft tracks (black lines) during the three NAAMES campaigns in November 2015, May 2016, and September 2017.

servations. NAAMES deployed the NASA C-130 aircraft to measure cloud and aerosol properties during three campaigns in November 2015, May 2016, and September 2017 over the approximate domain of 50–35° W, 38–60° N (Fig. 1).

Both NAAMES airborne in situ and remotely sensed observations are used to evaluate satellite retrievals of liquid cloud effective radius and optical depth from the Thirteenth Geostationary Operational Environmental Satellite (GOES-13) and from the MODIS aboard Aqua and Terra. The cloud properties were derived using the algorithms developed for the Clouds and the Earth's Radiant Energy System (CERES). The NAAMES observational dataset comprises in situ cloud observations collected by a Cloud Droplet Probe (CDP) and a Cloud Imaging Probe (CIP), as well as remotely sensed retrievals from the NASA Goddard Institute for Space Studies (GISS) airborne Research Scanning Polarimeter (RSP). A special emphasis is placed on the inter-satellite differences and the role of pixel resolution and viewing geometry in accounting for the observed discrepancies.

## 2 Dataset

### 2.1 Airborne observation

The NAAMES domain, aircraft tracks, and the mean Aqua MODIS low cloud fraction (cloud tops < 3 km) are depicted in Fig. 1. The region features mean cloud fractions greater than 0.65, with the dominant presence of supercooled cloud tops during the cold months (Hu et al., 2010), and corroborated by NAAMES RSP data as the presence of a rainbow (observed in cloud tops with liquid droplets) was prevalent during the three deployments. While the approximate duration of a C-130 flight was 10 h, dedicated in-cloud sampling

lasted around 60–100 min per mission, between 09:00 to 15:00 LT, when the solar zenith angle ranged from 23 to 81° (mean solar zenith angle of 51°). Cloud sampling was limited to boundary layer liquid clouds with a mean cloud-top height of 1376 m ± 602 m (±SD) and base height of 770 ± 363 m.

Cloud droplet size distributions were sampled in situ with a CDP manufactured by Droplet Measurement Technologies (DMT, Inc., Boulder, CO). The CDP probe is a forward-scattering optical spectrometer that measures droplet sizes between 2 and 50 μm with bin widths of 1 and 2 μm for droplet diameters larger or smaller than 14 μm, respectively. A main source of uncertainty is the oversizing and under-counting of droplet concentrations higher than 400 cm$^{-3}$ (Lance et al., 2010). This issue has a limited effect for NAAMES as the liquid cloud droplet number concentration remained below 250 cm$^{-3}$ during the campaigns. Unfortunately, post-deployment evaluation at DMT revealed that the probe operated with a sampling area larger than the manufacturer specifications, yielding an overcounting of droplets for all the bins. This overcounting is thought to equally affect each bin, implying that the cloud effective radius is little affected by the sampling area problem. Considering the unresolved problem with the CDP probe, cloud effective radius is the only in situ cloud observation used for quantitative assessments in this study. The error introduced by the larger instrument sample area in other quantities (water content, extinction coefficient, and cloud droplet number concentration) requires further analysis that will be undertaken in a future study. Large droplet sizes were sampled with the DMT CIP, which features 62 sizing bins with center sizes between 50 and 1600 μm (1.6 mm) and a width of 25 μm. Due to the more limited CIP sampling relative to other instruments (50 full profiles), we only use CIP data to infer the precipitation contribution to the total in situ cloud effective radius and how this could affect the analysis interpretation.

The determination of $r_e$ and $\tau$ used flight level ($\sim$ 7 km a.s.l.) solar polarized and unpolarized reflectance measurements taken at 0.865 μm from the airborne NASA GISS RSP while above cloud. Given the operational limitations of the CDP probe, RSP cloud products are a key dataset for this evaluation. The RSP features nine spectral bands between 410 and 2260 nm, with a field of view of 14 mrad, 14 mrad spacing between samples, and a scan swath of ±60° relative to nadir. The RSP polarimetric $r_e$ retrieval algorithm uses the polarized reflectance information contained in the backscattering angles ranging between 137 and 165° (Alexandrov et al., 2012). The retrieval method exploits the fact that the polarized reflectance in the rainbow angular range is well characterized by a unique combination of cloud effective radius and effective variance of the droplet size distribution. This justifies a parameterization that fits the angular shape of the polarized reflectance using an analytical equation dependent on scattering angle and scattering phase matrix, which in turn is calculated via Mie theory from $r_e$ and the effective variance of a gamma size distribution (Hansen

and Travis, 1974). A numerical assessment of the RSP using synthetic observations derived from a large-eddy simulation model quantifies an accuracy of RSP $r_e$ generally better than 0.15 μm, with retrievals insensitive to three-dimensional radiative transfer effects and $r_e$ representative of the microphysical properties of an optical depth within 1.0 from the cloud top (Alexandrov et al., 2012). In addition, analysis of RSP $r_e$ for NAAMES showed good quantitative agreement with CDP $r_e$ within about 100 m of the cloud top (Alexandrov et al., 2018). Unlike $r_e$, $\tau$ from the RSP is derived using a standard reflectance-based method that finds a value for $\tau$ that yields the best match between the observed 864 nm nadir reflectance and its simulated counterpart estimated with a one-dimensional radiative transfer model and constrained with the polarization-based $r_e$.

## 2.2 Satellite observations

The satellite cloud retrievals evaluated in this study are from GOES-13 and MODIS aboard Terra and Aqua. While GOES-13 observes the NAAMES domain continuously (GOES-13 is fixed relative to Earth and located at 75° W), Terra and Aqua daytime overpasses occur at approximately 10:30 and 13:30 local solar time (15:30 and 18:30 UTC), respectively. Cloud optical depth and effective radius are retrieved using CERES Edition 4 algorithms (Minnis et al., 2011, 2020) applied to MODIS using the 0.64 and 3.79 μm channels. CERES adopted these channels for $\tau$ and $r_e$ derivation because their radiometric equivalents are common to many other sun-synchronous and geostationary satellite imagers that are currently ingested by the CERES program. The CERES MODIS algorithms have been adapted to utilize similar channel combinations on geostationary (Minnis et al., 2008) and other lower Earth-orbiting satellites (Minnis et al., 2011, 2016) and integrated into the NASA Satellite ClOud and Radiation Property retrieval CE1 System (SatCORPS) to produce historical and near-real-time datasets for use in research and operations. Here, the Sat-CORPS uses the GOES-13 0.65 and 3.90 μm channels, with the visible radiances being calibrated against Aqua MODIS following Doelling et al. (2018). Lastly, we note that the algorithm for deriving satellite $r_e$ differs from the RSP algorithm, in that satellite-based $r_e$ relies on the dependence of shortwave-infrared unpolarized reflectance on $r_e$ (and an assumed value for effective variance, with near-infrared reflectance monotonically decreasing with $r_e$), whereas RSP is based on the dependence of the polarized reflectance on the scattering angle, $r_e$, and effective variance near the rainbow.

The SatCORPS team at NASA Langley provided near-real-time satellite support for the NAAMES operations (https://satcorps.larc.nasa.gov/NAAMES-2015, last access: 28 September 2021). This support included GOES-13 images and SatCORPS cloud retrievals every 30 min at a nadir resolution of 4 km (3.90 μm channel resolution and 0.63 μm channel subsampled to 4 km resolution). In practice,

given the high GOES viewing zenith angles ($\sim 65°$) for the NAAMES domain, the actual resolution for the GOES-13 imager is approximately $3.2\,\text{km} \times 9.3\,\text{km}$ for the east–west (zonal, 3.2 km) and meridional (9.3 km) components, respectively. To avoid retrievals with high uncertainties near twilight, we only use observations with solar zenith angles less than 75°. During NAAMES 2017, GOES-13 and GOES-16 took coincident measurements over the NAAMES domain, with GOES-16 ultimately replacing GOES-13 when it was decommissioned in December 2017. Due to calibration uncertainties prior to official implementation in NOAA operations, GOES-16 is not evaluated against NAAMES observations. However, we intercompare cloud products from the GOES-13 imager and the GOES-16 Advanced Baseline Imager (ABI) for December 2017 to provide a glimpse of improvements expected when using ABI data (Sect. 3.3). In addition to an increased number of channels, ABI features better spatial resolution (2 km at nadir for 3.90 μm) relative to its GOES-13 predecessor (4 km).

The MODIS cloud products evaluated here are identical to the ones used to generate the CERES Single Scanner Footprint (SSF) product. The SSF includes top-of-the-atmosphere radiative fluxes from the CERES instrument and MODIS cloud retrievals (CERES algorithm) averaged within the CERES footprint ($\sim 20\,\text{km}$, Loeb et al., 2018). Here, we use pixel resolution CERES MODIS retrievals ($1\,\text{km} \times 1\,\text{km}$ at nadir and $4.8 \times 2\,\text{km}$ at the scan edge) subsampled every other pixel, due to computational constraints, to achieve an effective $2\,\text{km} \times 2\,\text{km}$ resolution at nadir. Lastly, we note that the CERES cloud algorithms differ from those of the MODIS Science Team (Goddard Space Flight Center, Platnick et al., 2017). Even though both products compare well with each other, especially for low-level liquid clouds, some differences should be expected. The reader is referred to Painemal et al. (2012), Zhang et al. (2018), and Minnis et al. (2021) for a more in-depth comparison between the CERES and MODIS Science Team products.

## 2.3 Matching method

Collocation of satellite data and the aircraft observations are performed separately for the airborne in situ (CDP) and remotely sensed (RSP) data collection and depicted in Fig. 2.

### 2.3.1 Collocation with in situ data

Prior to matching the in situ and satellite data, we take into account that 3.79–3.9 μm satellite $r_e$ is representative of the first few optical depths ($\sim 2$) down from the cloud top (Platnick, 2000) where most of the absorption occurs for that band. Thus, this radiative signature implies that the $r_e$ comparison needs to be performed with in situ observations near the cloud top. For this purpose, we first estimate cloud boundaries (base and top) for continuous ascents and descent profiles by visually inspecting all the NAAMES in-cloud ob-

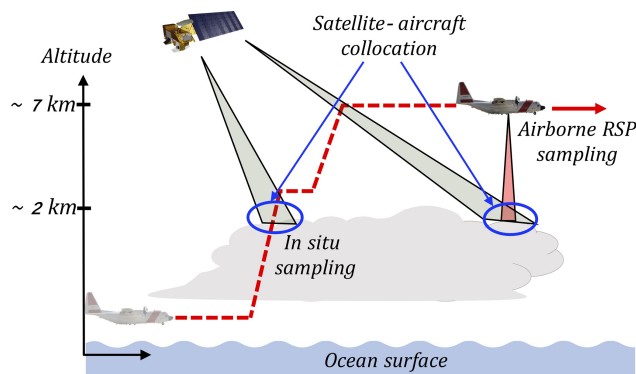

**Figure 2.** Collocation method between satellite and airborne in situ (CDP probe) and RSP observations. Satellite pixels are paired with in situ samples collected during profiling maneuver, whereas satellite and RSP data are collocated for high-altitude aircraft transects ($\sim 7\,\text{km}$).

servations and use a minimum liquid water content threshold of $0.03\,\text{g}\,\text{m}^{-3}$ to define a cloudy sample, a methodology that yields a total of 80 in situ samples. Next, cloud-top $r_e$ is computed for each profile by averaging $r_e$ over the uppermost portion of the cloud above the $\tau = 2.0$ altitude level from the top. In the calculation of $\tau$, we have assumed an extinction efficiency of 2.0, with cloud extinction coefficient estimated from the second moment of the droplet size distribution, as in Painemal and Zuidema (2011). The $r_e$ calculation is minimally sensitive to the $\tau$ threshold and CDP overcounting as variations of 1.0 and 3.0 (a range larger than CDP overcounting uncertainty) yield changes in $r_e$ close to 0.1 μm. Lastly, we match and average the closest $2 \times 2$ (GOES) and $4 \times 4$ (MODIS) pixels centered at the vertical profile location, with a temporal mismatch of less than 15 min for GOES-13 and 25 min for MODIS. The 25 min window for MODIS reflects the limited number of satellite overpasses available and represents a compromise between obtaining a meaningful number of collocated samples and ensuring that the aircraft and MODIS are observing the same cloud features.

### 2.3.2 Collocation with RSP

The two primary advantages of airborne RSP retrievals, relative to in-cloud CDP observations, are the increased spatiotemporal sampling and the satellite–RSP consistency in the sense that RSP $r_e$ is mostly sensitive to the cloud top ($\tau \sim 1$), similar to GOES and MODIS ($\tau \sim 2$, Platnick, 2000). Given the relatively narrow RSP field of view ($\sim 70\,\text{m}$ for NAAMES, Alexandrov et al., 2018), the RSP retrievals were averaged along the flight track to make it comparable to the satellite pixel resolution. Given an aircraft speed that ranges between $130–155\,\text{m}\,\text{s}^{-1}$ during the high-altitude aircraft transects (when RSP sampled boundary layer clouds), we use a 134 s average window, equivalent to a horizontal scale of at least 16 km. From the central latitude and longi-

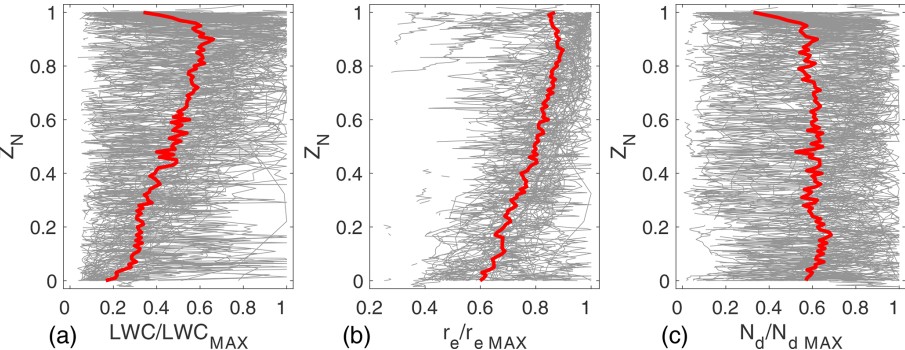

**Figure 3.** CDP profiles normalized by their maximum value from the three NAAMES campaign: **(a)** of liquid water content, **(b)** $r_e$, and **(c)** cloud droplet number concentration $N_d x$. Normalized height ($Z_N$) corresponds to 0.0 for cloud base and 1.0 for cloud-top height. Individual and mean profiles are depicted in gray and red, respectively.

tude of this window, a 16 km diameter allows for collocating around 2 north–south pixels for GOES-13 (4 subsampled pixels for MODIS) with the RSP retrievals, consistent with the methodology used for satellite–CDP collocation. As in the in situ collocation, the aircraft–satellite temporal mismatch is less than 15 and 25 min for the GOES imagers and MODIS, respectively. Consistency across the different analyses within this work indicates that the matching discrepancy between MODIS and GOES-13 has a negligible effect.

## 3 Results

### 3.1 Evaluations of satellite-derived cloud effective radius against CDP measurements

Before describing the main results, we first provide an overview of the cloud vertical structure during the campaign. The profiles in Fig. 3 are normalized by their maximum value, and cloud base and top are denoted, respectively, by 0 and 1 in the vertical coordinate ($Z_N$). Given the normalization applied to the data, uncertainties in the CDP should have a negligible impact in the result interpretation. The liquid water content profiles (Fig. 3a), on average, linearly increase with height until it sharply decreases at the cloud top. Similarly, $r_e$ linearly increases toward the cloud top, whereas $N_d$ is relatively homogeneous with height. While the vertical variability is substantial, the mean cloud structure observed in NAAMES is similar to that observed in more archetypal subtropical stratocumulus clouds (e.g., Painemal and Zuidema, 2011). Two key aspects that emerge from the normalized profiles are the following: (a) $r_e$ is a maximum near the cloud top, and (b) a vertically stratified cloud model is expected to fit the observations reasonably well, implying that liquid water path (LWP) can be more precisely estimated by LWP $= 5/9 \cdot \rho \cdot r_e \cdot \tau$ (with $\rho$ denoting the liquid water density), as opposed to the vertically homogeneous equation LWP $= 2/3 \cdot \rho \cdot r_e \cdot \tau$, as suggested by studies in the subtropics (e.g., Seethala and Horvath, 2010; Painemal et al., 2017).

Comparisons of satellite $r_e$ against its in situ counterpart (Fig. 4 and Table 1) reveal correlations of 0.68 for GOES and 0.58 for MODIS, with systematic positive biases. The overestimation by GOES reaches a value of 4.8 μm (45.7 %), which is more than twice that observed for MODIS (1.7 μm, 16.2 %). Similarly, the root mean square error (RMSE) is higher for GOES-13 (5.8 μm) than MODIS (2.9 μm). These findings are confirmed in the next section with the use of RSP data.

### 3.2 Evaluations of satellite-derived cloud effective radius and optical depth against RSP retrievals

The RSP–satellite $r_e$ linear correlation coefficient ($r$) is 0.52 for GOES and 0.68 for MODIS (Fig. 5, Table 2). A persistent positive bias is also confirmed for both satellite sensors, with values of 5.3 μm (51.6 %) for GOES and 2.60 μm (25.8 %) for MODIS (Table 2), which is slightly greater than those estimated from the CDP probe. The effect of spatial inhomogeneity in satellite $r_e$ was assessed by means of the $\tau$ coefficient of variation ($\chi$), determined as the ratio of the standard deviation to the mean RSP cloud optical depth (similar to Liang et al., 2009). The most heterogeneous samples, defined as the top $\chi$ quintile ($\chi > 0.8$, Fig. 5, filled blue circles), were contrasted against the rest of the samples. For GOES-13, comparing against heterogeneous samples ($\chi > 0.8$) yields a modest bias increase relative to samples with $\chi \leq 0.8$ (5.8 and 5.2 μm, respectively). Yet, the effects of heterogeneity on satellite $r_e$ are consistent with the overestimation that is expected for subpixel variability in cloud reflectances, although we note that the effects of heterogeneity are greatly ameliorated for $r_e$ retrievals estimated from the 3.7–3.9 μm band relative to those based on shorter wavelengths (Painemal et al., 2013).

We repeat the analysis above but applied it to $\tau$ (Fig. 6). The satellite and RSP $\tau$ yield higher linear correlation coefficients than those for $r_e$ ($r = 0.76$), with the satellite underestimating airborne $\tau$ by $-3.0$ for both GOES-13 and

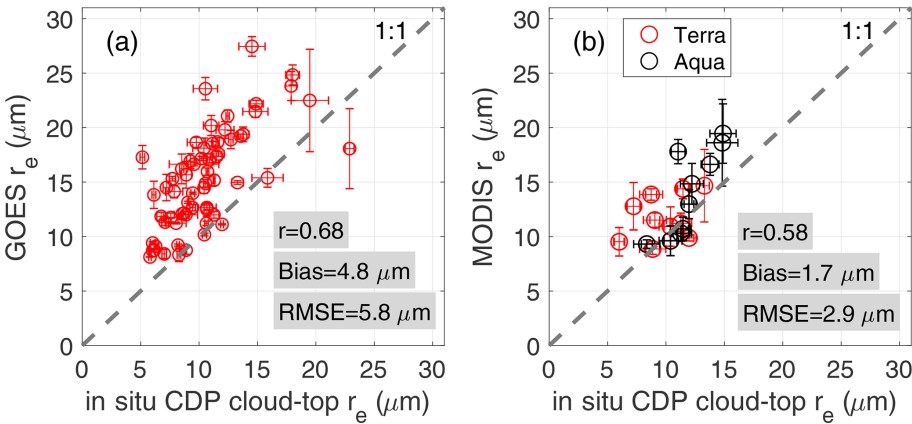

**Figure 4.** In situ CDP cloud-top effective radius against **(a)** GOES-13 and **(b)** MODIS. Linear correlation coefficient is denoted by $r$, bias is calculated relative to the in situ $r_e$, and RMSE is the root mean square error. Statistics for MODIS combine data from Aqua and Terra. Dashed line represents the 1-to-1 relationship.

**Table 1.** General $r_e$ statistics between satellite and CDP probe observations. Percentage values are relative to mean CDP values matched with the satellite data.

|  | GOES-13 vs. CDP | | | MODIS vs. CDP | | |
|---|---|---|---|---|---|---|
|  | Bias | $r$ | RMSE | Bias | $r$ | RMSE |
| $r_e$ | 4.8 µm (45.7 %) | 0.68 | 5.8 µm (55.3 %) | 1.7 µm (16.2 %) | 0.58 | 2.9 µm (27.7 %) |

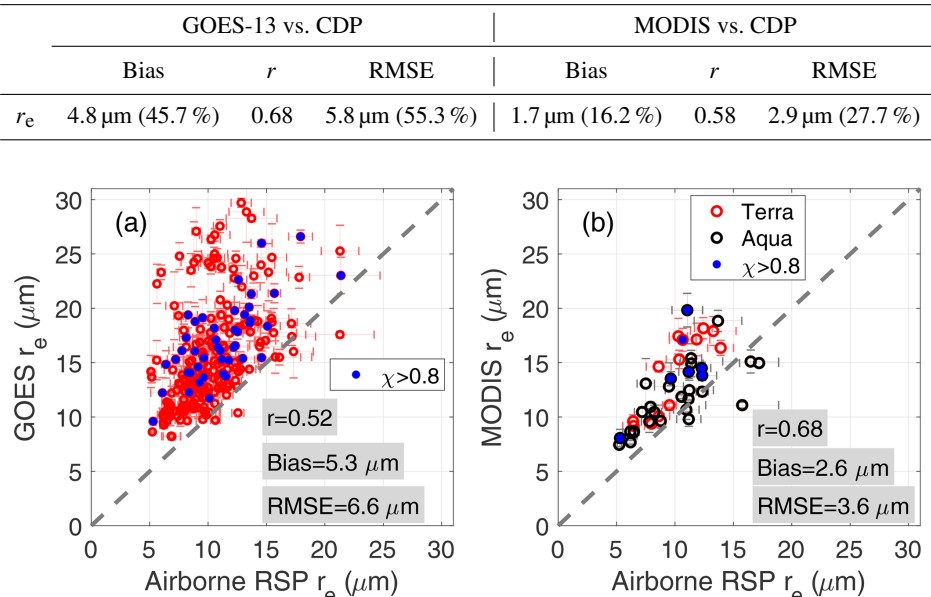

**Figure 5.** Relationship between airborne RSP $r_e$ and **(a)** GOES-13 and **(b)** MODIS. Statistics for MODIS combine data from Aqua and Terra. Error bars denote the spatial standard deviation. Blue circles denote retrievals derived over highly heterogeneous cloud scenes.

MODIS (Table 2). Unlike the $r_e$ comparison, the effect of the scene heterogeneity in satellite $\tau$ is evident, with negative biases reaching, respectively, $-10.9$ and $-8.2$ for GOES and MODIS for highly heterogeneous fields ($\chi > 0.8$). In contrast, more homogenous samples ($\chi \leq 0.8$) yield a reduced bias of $-1.2$ (GOES) and $-1.7$ (MODIS), which further decreases to $-0.72$ and $-0.20$ for scenes with $\chi < 0.5$.

### 3.3 GOES-13 and MODIS intercomparison

We further intercompare both satellite products to gain insight into the discrepancies between GOES and MODIS manifested in their different $r_e$ biases. We gridded the satellite data at 0.25° spatial resolution and matched them to within 15 min of the satellite overpasses for the NAAMES days over the oceanic domain bounded by 50–35° W, 40–60° N. The comparison for overcast grids shows that the GOES-13 $r_e$ is larger than both Terra MODIS (1.9 µm, Fig. 7a) and Aqua MODIS (2.0 µm, Fig. 8a) and the lin-

**Table 2.** General $r_e$ and $\tau$ statistics between satellite and RSP retrievals. Percentage values are relative to mean RSP values matched with the satellite data.

| | GOES-13 vs. RSP | | | MODIS vs. RSP | | |
| --- | --- | --- | --- | --- | --- | --- |
| | Bias | $r$ | RMSE | Bias | $r$ | RMSE |
| $r_e$ | 5.3 µm (51.6 %) | 0.52 | 6.6 µm (64.3 %) | 2.6 µm (25.8 %) | 0.68 | 3.6 µm (35.7 %) |
| $\tau$ | −3.0 (−20.8 %) | 0.76 | 8.4 (58.3 %) | −3.0 (20.3 %) | 0.77 | 7.4 (50.1 %) |

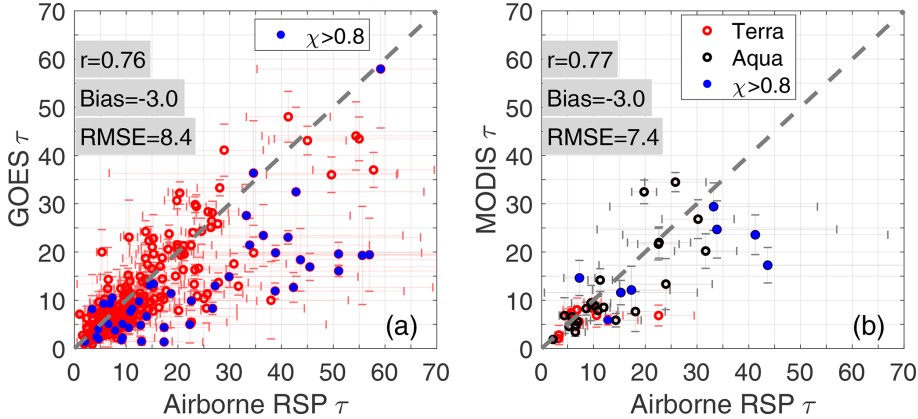

**Figure 6.** As Fig. 5 but for the comparison between airborne RSP $\tau$ and **(a)** GOES-13 and **(b)** MODIS.

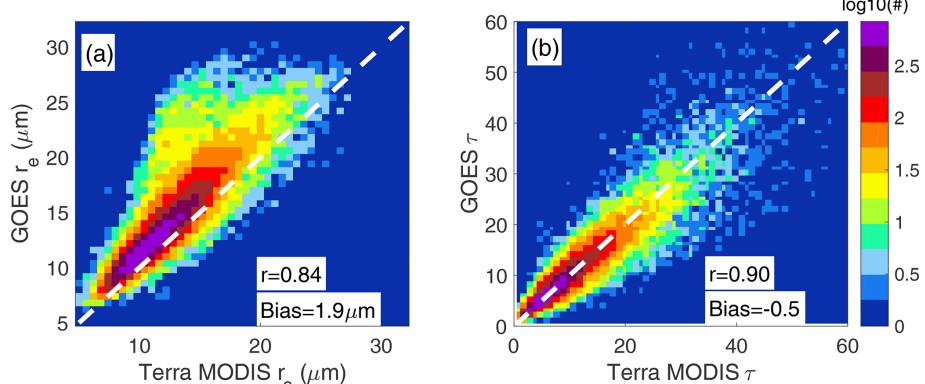

**Figure 7.** Bivariate histogram between Terra MODIS and GOES-13 **(a)** $r_e$ and **(b)** $\tau$. One-to-one line is denoted by the white dashed lines. Bias represents the mean difference between GOES-13 and MODIS.

ear correlations are $r = 0.84$–$0.90$. By contrast, GOES and MODIS $\tau$ values exhibit smaller differences ($< 0.7$), a smaller mean GOES $\tau$, and comparable correlations with $r = 0.90$ (Figs. 7b and 8b). Differences between the GOES and MODIS retrievals likely reflect (a) the fixed viewing geometry of GOES with an average viewing zenith angle of 64° and (b) higher MODIS pixel resolution. Both effects are illustrated in Fig. 9, in which the difference between GOES and MODIS (Terra and Aqua combined) cloud products are binned as a function of MODIS viewing zenith angle (VZA). Differences in $r_e$ decrease from nearly 2.2 µm near nadir to 1.5 µm close to the MODIS scan edge ($\sim 60°$, Fig. 9a). $\tau$

differences also decrease with MODIS VZA (within 1.2), with negligible GOES–MODIS difference for grids collocated near the MODIS scan edge. Despite closer agreement between GOES-13 and MODIS $r_e$ for high MODIS VZA, systematically larger GOES-13 $r_e$ than MODIS points to other factors in explaining the systematic biases for GOES-13.

While some aspects of the viewing geometry and illumination effects on MODIS $r_e$ and $\tau$ have been explored to some degree in a number of studies (e.g., Marshak et al., 2006; Kato et al., 2006; Horvath et al., 2014), it remains largely unknown to what extent previous analyses are ap-

https://doi.org/10.5194/amt-14-1-2021

Atmos. Meas. Tech., 14, 1–14, 2021

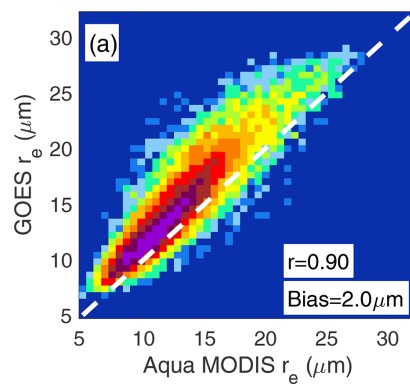 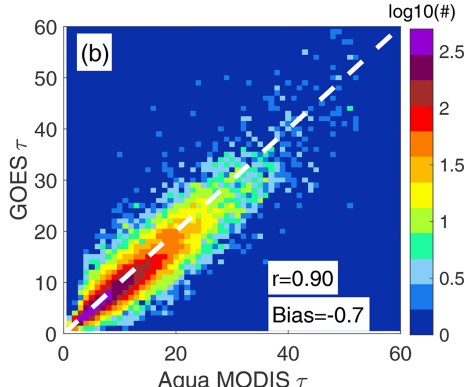

**Figure 8.** As Fig. 7 but for GOES-13 and Aqua MODIS.

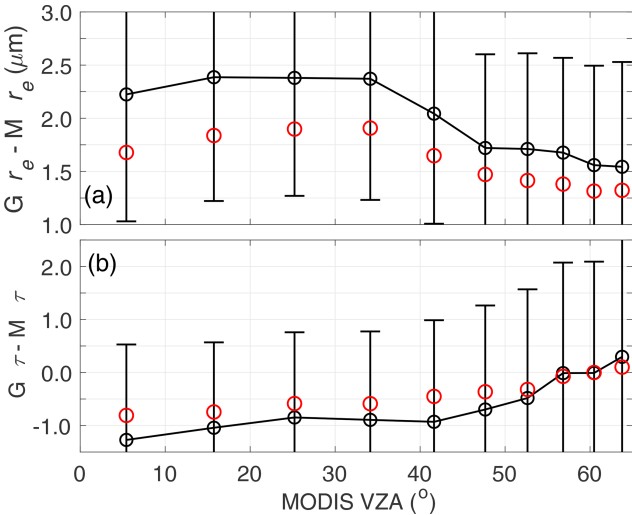

**Figure 9.** Mean differences between GOES and Aqua/Terra MODIS retrievals binned in MODIS VZA deciles: **(a)** cloud effective radius and **(b)** cloud optical depth. Error bars denote the standard deviation, and median values are represented by red circles.

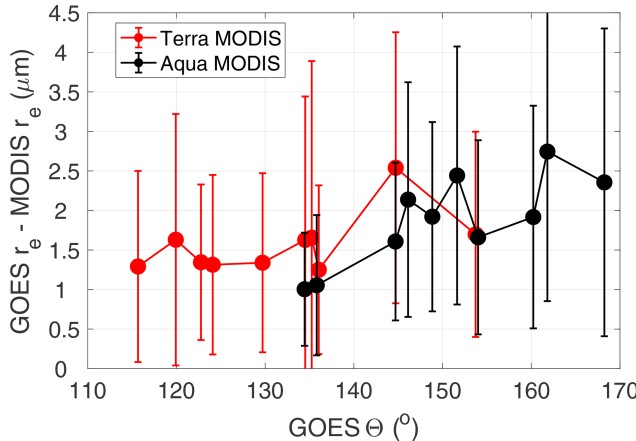

**Figure 10.** $r_e$ differences between GOES-13 and Terra MODIS (red) and Aqua MODIS (black) binned in deciles of GOES-13 scattering angle ($\Theta$). Error bars represent the root mean square difference for each bin.

plicable to geostatic sensor geometry. We have previously shown the sensitivity of the MODIS–GOES difference to VZA, consistent with the effect of pixel coarsening, and the non-linearity of the reflectance–$r_e$ and reflectance–$\tau$ relationship dependent on VZA (Liang and DiGirolamo, 2013). Another geometrical parameter of interest is the satellite scattering angle ($\Theta$), or the angle between the solar and satellite viewing direction, as it provides information about the cloud side viewed by the satellite (shadow or illuminated). For the data analyzed here, the GOES-13 grids matched with Terra and Aqua produce GOES $\Theta$ averages of 132.3 and 150.4°, respectively (mean scattering angles for Terra and Aqua MODIS are on average 127.5 and 122.7°, respectively). That is, GOES $\Theta$ in the afternoon is more oriented toward the backscattering direction. To examine the possibility of a bias dependence on $\Theta$, we bin the $r_e$ difference be-

tween GOES-13 and MODIS (Aqua and Terra) as a function of $\Theta$ (Fig. 10). It is found that the differences increase toward the backscattering direction for Aqua (black), particularly for angles higher than 140°, with changes of around 2.5 µm between the highest and lowest $\Theta$ bins. Differences between GOES-13 and Terra MODIS $r_e$ (Fig. 10, red) are small for $\Theta$ bins less than 135°, with GOES-13 $r_e$ larger than Terra MODIS for $\Theta = 143°$. Thus, this analysis suggests that overestimation in the GOES-13 $r_e$ increases for scattering angles greater than 140°. A similar analysis applied to MODIS $\Theta$ is more challenging because the range of MODIS $\Theta$ variability is narrower than GOES, and MODIS VZA and $\Theta$ cannot be fully disentangled.

## 4  Discussion

Since the comparisons were made using the cloud mode of the CDP particle size distribution, the potential effects of pre-

cipitation unaccounted for in the calculations are addressed here. This is because satellite retrievals can be positively biased relative to cloud mode observations under the presence of a precipitation mode not considered in the Mie calculations (Nakajima et al., 2010). The potential role of precipitation is indirectly assessed by comparing near-cloud-top CDP $r_e$ (cloud mode) and that derived from the CDP and CIP droplet size distribution, after discarding the first bin of the CIP probe (52 µm) to remove instrument sizing overlap. We found that total $r_e$ (CDP + CIP) is 0.41 µm larger than that from the CDP, a discrepancy that is much smaller than the difference between satellite and in situ $r_e$. This result is somewhat expected as precipitation tends to be weighted toward the cloud base, becoming an unlikely cause for satellite–aircraft discrepancy. It is possible that this effect becomes more relevant for shorter wavelengths, characterized by a deeper photon penetration into the cloud (e.g., 1.6 µm channel; Platnick, 2000). However, determining the extent of the precipitation-driven bias for other satellite channels is difficult, as $r_e$ estimated from shorter infrared wavelengths is more prone to subpixel variability and 3D radiative effects, which also yield particle size overestimations (e.g., Zhang et al., 2012).

Satellite values exceeding their RSP counterparts reflect in part the different sensitivity of each method to the cloud-top layer. For instance, in situ vertical profiles in Fig. 3 show a slight $r_e$ decrease at the cloud top. Because RSP $r_e$ is more sensitive to the optically thinner layer from the cloud top than those estimated from passive 3.7 and 3.9 µm channels, it is expected that even for unbiased retrievals, satellite $r_e$ would be larger than RSP $r_e$. However, this discrepancy should be modest as CDP $r_e$ averaged over an optical depth of 0.4 from the cloud top is only 0.17 µm smaller than that calculated for an optical depth of 2.0. The modest impact of the cloud vertical structure in explaining polarimetric and bi-spectral $r_e$ differences is also supported by 1D theoretical results in Miller et al. (2018) for retrievals derived at the same pixel resolution.

The effect of spatial resolution on the GOES-13 retrievals is explored by comparing GOES-13 and GOES-16 ABI for 5 d in December 2017, 2 months after the last NAAMES aircraft deployment. During December, both GOES satellites operated over the same region, implying nearly identical viewing geometries. With GOES-16 becoming the operational GOES-16, GOES-13 drifted to reach its final location at 60° E in January 2018. ABI pixel resolution is $2 \times 2$ km at nadir and $2.8 \times 4.6$ km for the NAAMES region, whereas the resolution of the GOES-13 imager is $3.1 \times 9.3$ km. GOES-16 and GOES-13 cloud products are retrieved with a very similar algorithm, with visible channels calibrated against Aqua MODIS, and, therefore, any inter-satellite discrepancy should be primarily attributed to the imagers' spatial resolution. Bivariate histograms of 0.25° averaged grids from GOES-16 and GOES-13 for the NAAMES domain are depicted in Fig. 11. GOES-13 $r_e$ is well corre-

lated with GOES-16 ($r = 0.97$), with GOES-13 sizes 0.7 µm larger than GOES-16 (Fig. 11a). A similar analysis applied to $\tau$ produced comparable correlations ($r = 0.93$), with GOES-13 $\tau$ being 3.4 less than that for GOES-16 (Fig. 11b). The $\tau$ negative bias systematically increases with GOES-16 $\tau$, with differences of $-1.1$ and $-5.8$ for GOES-16 $\tau$ of less than and more than 20, respectively. We note that December was characterized by optically thicker clouds than those observed during NAAMES, possibly attributed to the presence of low clouds driven by winter midlatitude weather disturbances. The observed inter-satellite differences are consistent with the effect of subpixel variability and the non-linearity between reflectance and $r_e$ and $\tau$. As the pixel resolution is degraded, the concave shape of the reflectivity–$\tau$ curve yields a retrieved $\tau$ from the pixel reflectance that is smaller than the average $\tau$ for that pixel, further explaining a positive bias in satellite $r_e$ due to the non-orthogonal relationship between $\tau$ and $r_e$. (see Fig. 1 in Marshak et al., 2006, and Zhang et al., 2012). Larger negative biases in $\tau$ as $\tau$ increases also appear to be linked to the concavity relationship in which the non-linear $\tau$–reflectivity relationship means that $\tau$ errors are accentuated for higher reflectances.

An additional factor known to severely affect plane-parallel cloud retrievals is 3D radiative transfer effects. While their influence is generally attenuated as the spatial averaging increases (pixel resolution coarsening, Marshak et al., 2006), for a specific combination of viewing angle, illumination, and cloud morphology, satellite-derived optical properties can be severely biased. This issue has been partially addressed here by examining the dependence on satellite scattering angle, which is generally assumed to provide information regarding cloud shadowing for the forward-scattering view ($\Theta < 90$) and enhanced illumination for the backscattering directions ($\Theta > 90$). Under this simple framework, it is generally interpreted that high values of reflectance at the backscattering direction are associated with an overestimation of $\tau$ and underestimation of $r_e$, and vice versa for the forward-scattering direction where shadowing occurs (Kato et al., 2006). Indeed, MODIS observations over Brazil have shown differences between forward and backscattering angles up to 6 µm for $r_e$ for cumulus clouds (Vant-Hull et al., 2007). Surprisingly, we found instead that GOES $r_e$ increases in the backscattering direction, reaching a cloud effective radius at $\Theta \geq 148°$ that is between 0.3–1.0 µm greater than that for $\Theta = 116°$. This small $r_e$ increase with $\Theta$ in the backscattering direction was also observed by McHardy et al. (2018) over the continental United States for GOES East and West. Moreover, they found that the expected increase in $r_e$ due to cloud shadowing (forward scattering) is only apparent for $\Theta < 90°$, with an increase greater than 10 µm for $\Theta = 60°$.

The positive bias in GOES-13 $r_e$ for the backscattering direction is somewhat consistent with other studies that report a modest MODIS $r_e$ increase over specific oceanic regions (e.g., Horvath et al., 2014; Liang et al., 2015). However,

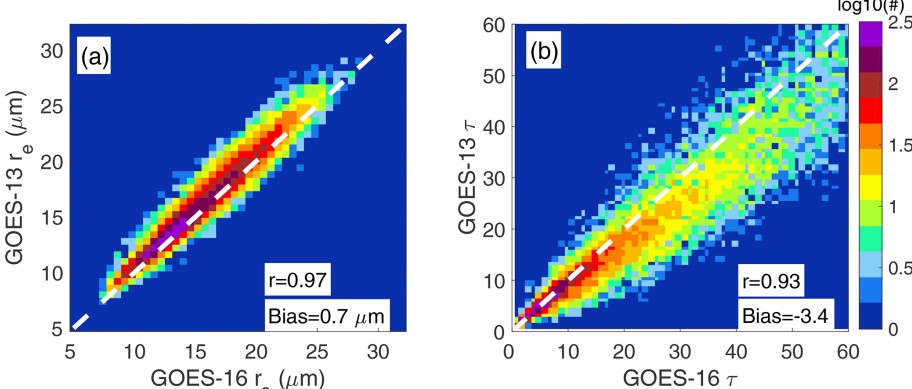

**Figure 11.** Relationship between 30 min GOES-16 ABI and GOES-13 cloud retrievals for 5 d of December 2017 (20, 22, 24, 26, 28) over the North Atlantic for solar zenith angle $< 75°$: **(a)** cloud effective radius and **(b)** cloud optical depth. The bias is defined as the mean difference between GOES-13 and GOES-16 retrievals.

since $\Theta$, latitudinal location, and viewing zenith angle are not decoupled in the MODIS data, isolating the effect of satellite scattering angle on MODIS retrievals is a challenge. Unlike MODIS, a wide range of scattering angles can be readily sampled by geostationary sensors. For instance Arduini et al. (2005) found for angles in the vicinity of the rainbow scattering angle ($\sim 140°$) a strong dependence of GOES $r_e$ on the prescribed effective variance of the droplet size distribution and a limited effect on $\tau$. Building upon Arduini et al. (2005), Benas et al. (2019) retrieved $\tau$ and $r_e$ from the Spinning Enhanced Visible and Infrared Imager (SEVIRI), aboard Meteosat 8 and 10, for a set of effective variances over the southeast Atlantic Ocean. They found that increasing the effective variance in the algorithm yields larger $r_e$ near the rainbow and smaller $r_e$ at the glory. Moreover, Benas et al. (2019) also noted that small effective variances tend to produce a more homogeneous diurnal cycle by reducing local discontinuities for the glory and rainbow angles. The exploratory analyses of Arduini et al. (2005) and Benas et al. (2019) leave, nevertheless, several unaddressed aspects such as the role of solar zenith angle, the contribution of 3D radiative effects, and the dependence of $r_e$ on the effective variance for a broad $r_e$ range. Currently, the CERES and SatCORPS cloud algorithms use a cloud model with a gamma-distribution effective variance of 0.1, which is higher than those observed over the ocean and less than those over land according to the literature review in Benas et al. (2019). For the nearly 80 profiles used in this study, we calculate the effective variance ($v_{eff}$) from the CDP probe as

$$v_{eff} = \frac{\int_0^{r_{max}} (r - r_e)\, r^2 n(r)\, \mathrm{d}r}{r_e^2 \int_0^{r_{max}} r^2 n(r)\, \mathrm{d}r}, \tag{1}$$

with $n(r)$, $r$, and $r_{max}$ denoting, respectively, the droplet size distribution, droplet radius, and maximum droplet radius in the distribution. We confirm that the effective variance is typically less than 0.1 (Fig. 12), with a mean value of 0.05 near

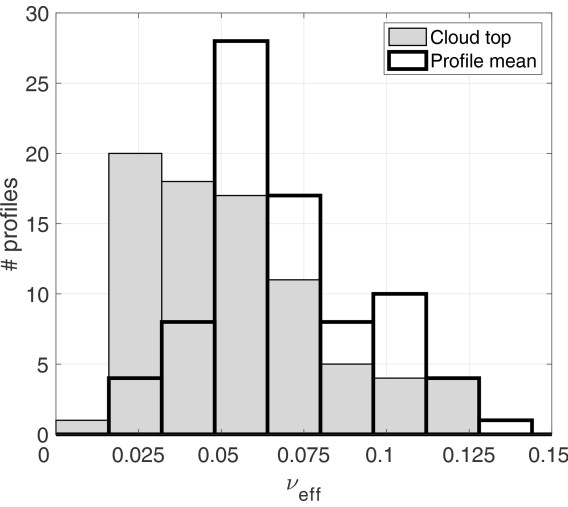

**Figure 12.** Histogram of computed effective variance ($\nu_{eff}$) assuming a modified gamma distribution for CDP values sampled near the cloud top (filled gray) and averaged throughout the cloud profile (black).

the cloud top and 0.07 for the averaged cloud profiles. Future work will concentrate on deriving cloud retrievals based on Mie calculations estimated using various droplet size distribution shapes to scrutinize their effect on effective radius biases in the backscattering direction.

In addition to pixel resolution and viewing geometry differences, the dissimilar spectral response between MODIS and the GOES-13 imager could yield retrieval discrepancies if the sensor differences are not properly accounted for in the algorithm, especially considering the spectrally wider GOES-13 channel. To circumvent this problem, rather than deriving optical properties for the central wavelength, we derive solar reflectances (lookup tables) for GOES-13 using weighted-average optical properties based on the instru-

ment's spectral response function. An aspect more difficult to address is the retrieval dependence on the index of refraction dataset. Platnick et al. (2020) found that retrieval differences that arise from the choice of refractive index dataset could explain $r_e$ differences between MODIS and the Visible Infrared Imaging Radiometer Suite (VIIRS) on Suomi NPP over ocean of about 1 μm for the 3.7 μm band. While the use of a specific refractive index dataset needs to be scrutinized, we note that pixel resolution and viewing zenith angle (Figs. 9 and 11a) could explain most of the 2 μm bias of $r_e$ GOES-13 relative to that of MODIS.

## 5   Concluding remarks

Airborne observations of cloud microphysical/optical properties of North Atlantic boundary layer clouds during NAAMES provided a suitable dataset for assessing cloud retrievals from GOES-13 and Terra/Aqua MODIS. The airborne dataset consists of in situ $r_e$ derived from the DMT CDP cloud probe, as well as retrievals of $\tau$ and $r_e$ from NASA GISS RSP measurements. The polarimetric $r_e$ retrievals from the RSP are largely insensitive to 3D radiative effects. This study provides one of the first satellite evaluations in midlatitudes poleward of 40°, where both warm and supercooled boundary layer clouds are a climatological feature. Our main findings are summarized as follows.

1. Comparisons between GOES-13 and MODIS $r_e$ and $\tau$ against airborne observations show good correlations: $r \geq 0.52$ for $r_e$ and $r \geq 0.76$ for $\tau$. Both satellite sensors yield positive $r_e$ biases relative to the airborne data. The GOES-13 bias exceeds that of MODIS by at least 1.9 μm. The positive MODIS $r_e$ bias is similar to, if not slightly higher than, that observed over the subtropical southeast Pacific in Painemal and Zuidema (2011). The GOES-13 and MODIS retrievals underestimate the RSP $\tau$ by 3.0, a difference primarily explained by subpixel heterogeneity, in which retrievals for pixels in spatially heterogeneous cloud fields are less than the expected mean $\tau$ for the same pixels. In contrast, spatial inhomogeneity effects have a modest effect on $r_e$, consistent with the weak sensitivity of the 3.79–3.9 μm band to spatial inhomogeneity (Zhang and Platnick, 2011).

2. Part of the large GOES-13 $r_e$ bias is caused by the high viewing zenith angle ($\sim 60°$) and the associated pixel coarsening. This effect is clearly observed when comparing GOES with MODIS for varying MODIS VZA. $r_e$ differences range from 2.2 μm for MODIS near-nadir view to 1.5 μm for a MODIS VZA similar to GOES VZA over the NAAMES region ($\sim 60°$). However, the discrepancy between GOES-13 and MODIS $r_e$ is not completely removed, and, thus, the GOES bias with respect to NAAMES observations remains high.

3. Pixel resolution effects are evaluated by comparing GOES-13 with GOES-16 when both satellites were situated close to each other, before GOES-13 drifted to its 60° W position. We find that GOES-13 $r_e$ is 0.7 μm larger than that from GOES-16. This difference is associated with a pixel area that decreases from 29.3 (GOES-13) to 12.9 km$^2$ (GOES-16). It is concluded that GOES-16 should yield a better agreement with ground-truth data, yet the satellite $r_e$ overestimation is not removed.

4. Exploratory analysis is intended to determine the impact of satellite scattering angle $\Theta$ on $r_e$. GOES-13 $r_e$ increases with $\Theta$ up to 1.0 μm relative to MODIS. The result is counterintuitive as the backscattering direction is expected to be associated with $r_e$ underestimation as the sensor views the bright side of the cloud. We lack a definitive explanation for the $\Theta$–$r_e$ relationship, and, thus, future work will address this with the use of a larger satellite dataset. Lastly, although GOES biases attributed to backscattering direction, high VZA, and pixel resolution might not be exactly additive, their magnitudes could well explain the discrepancy between GOES and MODIS.

Our assessment confirms some results in Ahn et al. (2018) and Kang et al. (2021) over the Southern Ocean, which were, to the best of our knowledge, the only MODIS assessments at high latitudes over the ocean based on in situ aircraft. Clouds reported in Ahn et al. (2018) correspond to highly broken stratocumulus clouds, which pose challenging conditions for both airborne sampling and satellite remote sensing. Even though they found a positive bias in MODIS $r_e$ (Goddard Space Flight Center level 2 product) for non-precipitating clouds, their limited dataset prevented an in-depth analysis of the reasons for the overestimation. In contrast, Kang et al. (2021) found linear correlation coefficients $\geq 0.78$ between CERES MODIS cloud retrievals ($r_e$, $\tau$, and LWP) and in situ cloud probes, with a positive bias of 1.5 μm for non- and lightly precipitating clouds in summer. Our findings are consistent with Kang et al. (2021) and other studies over the eastern Pacific, in which MODIS and GOES retrievals correlate well with airborne data, with larger satellite $r_e$ relative to in situ $r_e$. On the other hand, Witte et al. (2018) found an insignificant bias of MODIS Collection 6 (MODIS Science Team retrievals) relative to in situ Phase Doppler Interferometer (PDI) observations over the subtropical eastern Pacific. While Witte et al. (2018) pointed to the importance of counting on in situ observations that fully capture the droplet size distribution, our study relies on two independent airborne datasets, lending confidence in the satellite assessment. While accounting for precipitation in the in situ observation would decrease the MODIS $r_e$ bias by 0.41 μm, the remaining discrepancy is possibly explained by a combination of viewing geometry and 3D radiative transfer effects (Kato et al., 2006).

While independent aircraft datasets corroborated the results for satellite $r_e$, assessment of $\tau$ was based only on comparisons with the RSP $\tau$, with no direct estimates of liquid water path (LWP). However, an indirect LWP comparison can be achieved by applying the relationship LWP $= 5/9 \cdot \rho \cdot r_e \cdot \tau$ to both RSP and satellite data (Sect. 3.1). The correlations between satellite and RSP LWP are high ($r = 0.67$ for GOES-13 and $r = 0.73$ for MODIS), with satellite LWP overestimating that from RSP by 17.0 and $9.5\,\mathrm{g\,m^{-2}}$ for GOES-13 and MODIS, respectively. The satellite overestimation is caused by the $r_e$ bias, which also explains the higher GOES-13 LWP bias compared to that for MODIS LWP.

Our analysis underscores less understood uncertainties in cloud retrievals from geostationary satellites caused by the fixed geometry and the broad range of viewing zenith and scattering angles not observed in MODIS. Future work will expand the analysis with a more comprehensive satellite dataset including intercomparisons between GOES-13/16 and Aqua/Terra as well as from other sun-synchronous satellites. Further, radiative simulations and the development of a geostationary simulator will be valuable for interpreting the observational relationships.

*Data availability.* NAAMES data are publicly available in the NASA Atmospheric Science Data Center (ASDC; https://doi.org/10.5067/Suborbital/NAAMES/DATA001, NASA, 2020). GOES-13 cloud retrievals are available at https://satcorps.larc.nasa.gov/prod/naames/prod-satp-netcdf/ (last access: 28 September 2021, NASA Langley Research Center, 2021), and the latest data for flight days are also available upon request.

*Author contributions.* DP designed the study, and DP and DS carried out the analysis. DP prepared the manuscript with contributions from all the co-authors. PM and WS helped with the interpretation of the satellite analysis. BC, EC, and RM provided their insight on the use of NAAMES in situ and remotely sensed observations. BC, RM, EC, CR, KT, EW, and LZ collected the airborne in situ and remotely sensed observational data during NAAMES.

*Competing interests.* The authors declare that they have no conflict of interest.

*Acknowledgements.* We thank the NAAMES aircraft team and PI Michael Behrenfeld for their tireless efforts during the 5-year project. GOES processing by Rabindra Palikonda is greatly appreciated.

*Financial support.* This work was funded by the NAAMES project and the CERES program. Richard H. Moore was partially supported by a NASA New Investigator (Early Career) Program award.

*Review statement.* This paper was edited by Simone Lolli and reviewed by three anonymous referees.

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

**Remarks from the language copy-editor**

CE1    I do not understand your comment. There was no website link here in your submitted manuscript. I simply added "retrieval" to the definition as in https://satcorps.larc.nasa.gov/ and asked you to confirm. If anything is missing here, then please clarify.

**Remarks from the typesetter**

TS1    Please note that value changes are not language changes and require editor approval before they can be inserted. If you still insist on changing this value, please give an explanation for the editor for why this needs to be changed. Thanks.

TS2    Please note that the URL has to be the same as the one cited in the "Data availability" section.