# Peer review of "Evaluation of satellite retrievals of liquid clouds from the GOES-13 Imager and MODIS over the midlatitude North Atlantic during NAAMES campaign"

_Atmospheric Measurement Techniques, 2021_

## Author Response (AR1)

**Response to Reviewer 1**

We sincerely appreciate the reviewer's suggestions and careful reading of the manuscript. Our responses to his/her comments are listed below (in blue):

1)        The authors state that the RSP $r_e$ retrievals are at 2.26 µm (line 105, page 4). I could not find information about retrievals at 2.26 µm using the polarimetric technique in the various references. The authors should give more details about the retrieval technique, and perhaps explain why this channel was chosen to retrieve $r_e$.

We appreciate the reviewer's help in finding this error in the RSP description. The reviewer is correct in that RSP $r_e$ is not derived using the 2.26- µm polarized channel. Instead, the retrieval is derived from the 0.865-µm polarized channel (see Alexandrov et al, 2012, section 3). We have rewritten the RSP description to read:

"Flight level ( ~ 7 km ASL) solar polarized and unpolarized reflectance measurements taken at 0.865 µm from the airborne NASA GISS RSP while above cloud were used to derive $r_e$ and $\tau$"

More details about the technique are provided in our response below.

2) How do the authors justify that the RSP $r_e$ retrievals at 2.26 µm are mostly sensitive to the cloud top (optical depth ≈ 1) (lines 176-177, page 6)?  Based on the work by Platnick (2000), they state that 3.79-3.9 µm satellite $r_e$ is representative of about 2 optical depths down from the cloud top (line 161, page 5). Stating that the RSP retrievals at 2.26 µm are weighted higher in the cloud than the satellite retrievals at 3.79-3.9 µm seems inconsistent with Platnick (2000). Please explain.

The RSP retrieving technique substantially differs from the standard satellite visible bi-spectral reflectance technique. That is, the photon penetration principle described in Platnick (2000) for MODIS-like retrievals is not applicable to the polarization technique. Briefly, The RSP views the same target at different angles around the rainbow scattering (~137-165˚), allowing for multiple polarized reflectance measurements for the same target (see Section 2.1). Thus, the algorithm for retrieving RSP $r_e$ exploits the shape of the polarized reflectance as a function of scattering angle. It has been shown that the shape of the polarized reflectance can be modeled as a function of scattering angle, cloud droplet effective radius, and effective variance of the droplet size distribution (via Mie calculations).  As the polarized rainbow is typically a single scattering phenomenon, this is generally limited to the uppermost cloud layer.  Numerical results in Alexandrov et al. (2012, in the reference section) confirm that RSP $r_e$ is sensitive to a cloud layer with an optical depth of around 1.0 from the cloud top. Regarding the use of RSP polarized reflectance based from other channels, differences in the retrieved $r_e$ are within 0.6 µm (Alexandrov et al., 2012), with differences primary influenced by the strength of the polarization signature, which tends to be weaker for longer wavelengths. In the revised manuscript, we corrected the description of the RSP algorithm.

3) I believe that a discussion regarding the expected differences between retrievals at 2.26 µm (RSP) and 3.70-3.90 µm (satellites) is missing.

As discussed in our previous response, the physical principle that enables the derivation of $r_e$ using RSP is different from the bi-spectral method applied to satellite imager observations. We have added the following sentence to emphasize the different technique.

" Lastly, we note that the algorithm for deriving satellite $r_e$ differs from the RSP algorithm, in that satellite-based $r_e$ relies on the dependence of shortwave-infrared unpolarized reflectance on $r_e$ (and an assumed value for effective variance, with reflectance monotonically decreasing with $r_e$), whereas RSP is based on the dependence of the polarized reflectance on the scattering angle, $r_e$, and effective variance near the rainbow."

4) The authors state that post-deployment evaluation of the CDP probe showed that there was an overcounting of droplets for all bins (lines 94-96, page 3) and that as a result, CDP could provide only $r_e$, but not water content, extinction coefficient, and cloud droplet number concentration. Therefore, I don't understand how the authors could determine the $\tau = 2$ altitude level from the top to determine CDP cloud-top $r_e$ (lines 166-167, page 5). Please explain.

The reviewer raises a good point. Uncertainties in the sampling area are expected to be much less than 50 %. This uncertainty would propagate to cloud droplet number concentration as well as extinction and water content, yielding similar relative biases. We tested $\tau = 1$-3 (a range that is larger than the expected uncertainty in the measurements) and the results were nearly identical for cloud droplet effective radius ($\pm 0.1$ µm). In other words, the optical depth threshold is insensitive to the overcounting issue with the CDP probe. We have added the following sentence to explain this point:

"The $r_e$ calculation is minimally sensitive to the $\tau$ threshold and CDP overcounting as variations of 1.0 and 3.0 (a range larger than CDP overcounting uncertainty) yield changes in $r_e$ close to 0.1 µm"

5) The authors mention the presence of supercooled liquid water clouds during the cold months when they present the airborne observations (line 85, page 3), and in the conclusion, they state that both supercooled and warm boundary layer clouds are a climatological feature (lines 331-332, page 10). However, I do not see any discussion on this topic in the presentation of the results. Why is this important? Are the comparisons different for supercooled and water clouds? Please develop.

Given the increasing interest in supercooled clouds and their importance in climate sensitivity simulated by models (e.g. Zelinka et al. 2020, GRL, https://doi.org/10.1029/2019GL085782), we briefly mentioned the supercooled cloud presence during periods of the NAAMES deployment. From a remote sensing perspective, supercooled liquid and warm clouds are treated in the same by the algorithm in terms of their optical properties.

Specific comments
Abstract, line 18: I suggest specifying in the abstract which GOES-13 and MODIS (Aqua and Terra) products are used for the study.
Done, thanks

"Satellite retrievals of cloud droplet effective radius ($r_e$) and optical depth ($\tau$) from the Thirteenth Geostationary Operational Environmental Satellite (GOES-13), and the MOderate resolution Imaging Spectroradiometer (MODIS) onboard Aqua and Terra, based on the Cloud and the Earth's Radiant Energy System project algorithm,.."

Introduction or where relevant: please define "effective radius".

We added the following description in the introduction:

"cloud effective radius ($r_e$, the ratio of the third to the second moment of the droplet size distribution)"

Lines 70 and 84, page 3: I found that Fig. 1 was not very informative because too small. I would suggest 1 panel per campaign with the associated mean Aqua "cloud cover". It looks like the caption should actually say "low cloud fraction".

We modified the figure following the recommendations of the reviewer. Given the main focus of the paper, assessment of satellite retrievals for overcast scenes, a detailed description of cloud variability during the three NAAMES campaigns is beyond the scope of our study.

[Figure]

Figure 1: Mean Aqua-MODIS low-cloud cover and aircraft tracks (black lines) during the three NAAMES campaigns in November 2015, May 2016, and September 2017.

Line 85, page 3: please explain how RSP data could confirm the presence of supercooled cloud tops during the cold months. During which campaign(s)?

Supercooled clouds typically occurred during the November campaign (2015). As the rainbow is a signature of liquid droplets, as long as the cloud temperature is below 0˚C and the RSP observes the rainbow, the cloud can be identified as being formed by supercooled clouds at the cloud top. We added the following sentence to explain this:

"…and corroborated by NAAMES RSP data as the presence of a rainbow (observed in cloud tops with liquid droplets) was prevalent during the three deployments."

Line 150, page 5: Please describe the CERES SSF product and provide a reference.

In the revised version, we added the following sentence:

"The MODIS cloud products evaluated here are identical to the ones used to generate the CERES Single Scanner Footprint (SSF) product. SSF includes top-of-the-atmosphere radiative fluxes from the CERES instrument and MODIS cloud retrievals (CERES algorithm) averaged within the CERES footprint (~20 km, Loeb et al., 2018). Here, we use pixel resolution CERES-

MODIS retrievals (1 km x 1 km at nadir and 4.8x2 km at the scan edge) subsampled every other pixel, due to computational constraints, to achieve an effective 2 km x 2 km resolution at nadir."

Line 193, page 6: The authors give an overview of the cloud vertical structure during the campaign. Is it for only one of the 3 campaigns? If yes, which one? Please clarify. Should results for warm clouds and supercooled clouds be shown separately?
Again,
The reviewer is correct, the profiles were constructed using data from the three campaigns (this is clarified in the revised version). Since the occurrence of supercooled and warm liquid clouds is not relevant from a remote sensing perspective, we decide to not include such analysis.

Line 195, page6; Fig. 3: the authors state earlier in the text (lines 86-87) that cloud sampling was limited to boundary layer liquid clouds with a mean cloud top height of 1376 m±602 m (± standard deviation). What about cloud base?
770± 363 m. The value is now reported in the revised manuscript.
Lines 287 to 290, page 9: this discussion is difficult to follow without an illustration or at least a reference.
Rather than repeating schematics that are available in the literature, we now cite Figure 1 in Marshak et al. (2006, in the reference section).

Lines 321-322, page 10: please explain how the effective variances shown in Fig. 12 (not Fig.11) were retrieved. Is it from RSP or CDP?
Figure 12 was derived from the CDP probe using the following formula:

$$v_{eff} = \frac{\int_0^{r_{max}} (r - r_e) r^2 n(r) dr}{r_e^2 \int_0^{r_{max}} r^2 n(r) dr}$$

The definition of effective variance is now included in the manuscript.
Technical comments:
Line 140, page 5: "east-west" => could be rephrased.
We slightly rephrased the sentence to read:
"Imager is approximately 3.2 km x 9.3 km for the east-west (zonal, 3.2 km) and meridional (9.3 km) components, respectively."

Line 321-322, page 10: Fig.11 should be Fig. 12
Corrected, thanks.

References: the format does not seem compliant with AMT specifications.
We appreciate the reviewer's comments. The revised manuscript lists the references with the appropriate format.

**Response to Reviewer 2**

We appreciate the reviewer's thoughtful comments and key suggestions. Our responses to his/her comments are highlighted in blue:

- Page 2, lines 56-57: I suggest citing the Witte et al. (2018) paper here, along with implications on this study. (Witte, M. K., Yuan, T., Chuang, P. Y., Platnick, S., Meyer, K. G., Wind, G., & Jonsson, H. H. (2018). MODIS Retrievals of Cloud Effective Radius in Marine Stratocumulus Exhibit No Significant Bias. Geophysical Research Letters, 45(19), 10,656–10,664, doi:10.1029/2018GL079325.)

Thank you for drawing our attention to Witte et al., which presents an interesting perspective on satellite retrieval biases dependent on the cloud probe. In the revised paper, we added the following sentence:

"On the other hand, Witte et al. (2018) found an insignificant bias of MODIS Collection 6 (MODIS Science Team retrievals) relative to in-situ Phase Doppler Interferometer (PDI) observations over the subtropical eastern Pacific. While Witte et al. (2018) point to the importance of counting on in-situ observation that fully capture the droplet size distribution, our study relies on two independent airborne datasets, lending confidence in the satellite assessment."

- Page 3, line 86: Just to clarify, the C-130 only flew 1 to 1.5 hour (60 to 100 minute) flights? Seems short to me, so perhaps I'm misunderstanding.

The 1-1.5h flight duration mainly refers to the period of active in-cloud sampling, however, the total flight duration was 10 hour. NAAMES sampling also includes cloud-free and subcloud observations. Furthermore, the 10 hour flight also includes 4 hours transit. This information was updated in the revised manuscript.

- Page 3, lines 96-97: This statement on overcounting being thought to equally affect all size bins is an assertion without evidence. Was this verified to be the case? It might be an acceptable assumption, but there's nothing here that makes that case.

This operation is typical of the operation of optical probes, in which the concentration of a specific bin is equivalent Conc_bin = Counts_bin / (Sample Area) / (True Air Speed) (Brenguier et al., 1994). Thus, the sample area equally affects each bin of the probe, implying that cloud effective radius remains unaltered.

Brenguier, J. L., Baumgardner, D., & Baker, B. (1994). A Review and Discussion of Processing Algorithms for FSSP Concentration Measurements, *Journal of Atmospheric and Oceanic Technology*, *11*(5), 1409-1414.

- Page 3 line 102: Why is the CIP sampling not the same as the CDP? My understanding is that these two in situ instruments are supposed to complement each other to resolve the full width of the droplet size distribution, and are flown together for that reason.

We agree with the reviewer that, ideally, both cloud probes should be operated in tandem. Unfortunately, due to issues with the CIP measurements, a subset of CIP observations were deemed unreliable.

- Page 4, lines 115-117: Yes, the RSP polarimetric re retrievals may be accurate, at least for the synthetic LES cases considered in Alexandrov. But the key question here, given their use as a benchmark for satellite re retrievals, is whether these retrievals should be consistent with those from total reflectance approaches considering their different vertical weighting functions (e.g., Platnick (2000)). The polarimetric signal is a single-scattering phenomenon and thus is sensitive to the very top of the cloud. Looking at the re profiles in Fig. 3 (and from knowledge of similar profiles from other field campaigns), there is a decrease in re at the very top of the cloud. This decrease may in fact be too small to matter, but the authors don't fully address this other than later in the paper stating that using different tau thresholds (1 and 3) in their averaging of "cloud top" CDP measurements yields only a roughly 0.1μm re change. The single-scattering polarimetric signal may be in large part from the portions of the cloud above even 1 optical depth into the cloud. Please comment on this.

We have added the new sentence in the revised manuscript:

"Satellite $r_e$ larger than its RSP counterpart reflects in part the different sensitivity of each method to the cloud top layer. For instance, in-situ vertical profiles in Figure 3 shows a slight decrease in $r_e$ at the cloud top. Because RSP $r_e$ is more sensitive to the optically thinner layer from the cloud top than those estimated from passive 3.7-μm and 3.9-μm channels, it is expected that even for unbiased retrievals, satellite $r_e$ would be larger than RPS $r_e$. However, this discrepancy should be modest as CDP $r_e$ averaged over an optical depth of 0.4 from the cloud top is only 0.17 μm smaller than that calculated for an optical depth of 2.0."

- Page 4, lines 118-121: Radiometric calibration, and relative radiometry between two imagers, can have a big impact on tau retrievals and their agreement between two sensors (see, e.g., Meyer K, Platnick S, Holz R, Dutcher S, Quinn G, Nagle F. Derivation of Shortwave Radiometric Adjustments for SNPP and NOAA-20 VIIRS for the NASA MODIS-VIIRS Continuity Cloud Products. Remote Sensing. 2020; 12(24):4096. https://doi.org/10.3390/rs12244096). While the tau retrieval agreement is quite good later in the paper, did the authors assess the relative radiometry between RSP and MODIS/GOES? It's possible that the good agreement is fortuitous and may be masking larger heterogeneity effects.

RSP is calibrated in the GSFC calibration facility pre- and post- mission, or the ARC calibration facility. In both cases the radiance calibration is traceable to the NIST irradiance standard and uses an integrating sphere as described in https://www.nist.gov/sites/default/files/documents/calibrations/sp250-20.pdf. We have found that RSP is radiometrically stable to within 2% over a period of 5 years, and RSP is thermally controlled to the same room temperature (20°C) at which it is calibrated. Moreover, the agreement between the GLAMR/SIRCUS detector based calibration and the older lamp/irradiance based calibration was generally within 2%, and for the key 865 nm band was within 1%. Figure 8 in McCorkel et al. (2016) shows that Landsat8 OLI and RSP agreed to within about 2% for window channels. Assessing radiometry between MODIS/GOES and RSP is not easy in the North Atlantic because RSP has a smaller footprint than MODIS/GOES and cloud variability would make it very

difficult to assess differences at a meaningful level of fidelity. However, Figure 10 of Vermonte et al. (2016) suggests that OLI and MODIS agree well radiometrically at least for the red and NIR bands.

McCorkel, J., Cairns, B., and Wasilewski, A.: Imager-to-radiometer in-flight cross calibration: RSP radiometric comparison with airborne and satellite sensors, Atmos. Meas. Tech., 9, 955–962, https://doi.org/10.5194/amt-9-955-2016, 2016

Vermote, E., Justice, C., Claverie, M., and Franch, B.: Pre- liminary analysis of the performance of the Landsat 8/OLI land surface reflectance product, Remote Sens. Environ., https://doi.org/10.1016/j.rse.2016.04.008, 2016.

- Page 6, lines 177-178: See my comment above on the vertical weighting functions of polarimetry versus total reflectance. I guess for 3.7/3.9μm, the difference in weighting with respect to polarimetry is reduced compared to, say, 1.6μm, but this is a little hand-wavy and there may still be differences.

We agree with the reviewer. Our previous statement was inaccurate. We revise sentence to read:

"…the satellite-RSP consistency in the sense that RSP $r_e$ is mostly sensitive to the cloud top ($\tau$~1), comparable to GOES and MODIS ($\tau$~ 2, Platnick; 2000)." Additional information is also provided in our response to reviewer's comment concerning lines 115-117.

- Page 6, lines 204-205 and Fig 4: I suggest adding error bars to this plot similar to those in Fig. 5. For the MODIS vs CDP plot, can you stratify these results by the MODIS 250m heterogeneity index (Liang et al. (2009), again similar to what is done in Fig. 5)? Also, what about sensitivity to the width (effective variance) of the observed droplet size distribution? The satellite retrievals are making an assumption on veff (later on defined as 0.1) – how do these results stratify as a function of divergence of that veff assumption from the observations? Veff can be calculated from the observed DSDs, so I suggest doing that analysis.

Figure was updated following the recommendation of the reviewer. Regarding the inhomogeneity index estimated from satellites, a disadvantage of such calculation is that GOES-13 imager and MODIS pixel resolution is dissimilar. This implies that heterogeneity indices are satellite dependent, which is not ideal. In contrast, RSP is advantageous for deriving a heterogeneity metric as the same index, at the same resolution (which is much higher than MODIS) can be applied to any satellite sensors, becoming a more absolute way of quantifying inhomogeneity. The inherent assumption is that the RSP sampling is statistically representative of the wider area viewed by the satellite sensors.

Concerning veff, it is a challenging analysis due to several reasons. First, the range of variability observed at the cloud top (with most of the samples with veff<0.1, Figure 12) was narrow and the number of matched GOES-RSP samples was insufficient for a robust statistical

calculation (see our response to the comment below). In addition, based on ongoing work, the dependencies are highly non-linear and vary with viewing geometry, solar zenith angle, and particle size. The issue is complicated enough to be addressed in a standalone work. While we agree with the reviewer about the scientific value of pursuing a more comprehensive analysis, this is left for future work.

- Page 7, lines 213-215: Using RSP to define the heterogeneity index only provides information in one direction, i.e., along the flight track. Both satellite imagers have footprints much larger than the width of the RSP footprint, so across-track heterogeneity may be missed. Using the MODIS 250m heterogeneity, as I suggest above, would be helpful. Also, following my previous comment, what is the veff retrieved by RSP for these comparisons? Are the RSP veff generally consistent with CDP, at least where the two can be reasonably compared? I see the RSP veff are shown later in Fig. 12, but there is no stratification of CER retrieval differences as a function of veff deviation similar to what was done for VZA and scattering angle, or even heterogeneity. Veff sensitivity should be a no-brainer to add here.

  While the fine resolution of the 250-m MODIS channel is well-suited for inhomogeneity calculations, the coarser resolution of GOES-13 (1 km) would yield inhomogeneities indices that are not comparable with its MODIS counterpart. Instead, as previously discussed, we decided to use RSP as it detects cloud features at much higher spatial resolution, and the calculation can be matched and applied to both GOES-13 and MODIS.

  Concerning veff, comparisons between and RSP and CDP droplet size distributions during NAAMES in Alexandrov et al. (2012) suggest that RSP and CDP veff are generally consistent. The scatterplot below depicts $r_e$ differences between GOES-13 and RSP as a function of the effective variance from the RSP. It is unclear from Figure S1 GOES biases are related to veff. However, we need to carry out a more comprehensive analysis to test the hypothesis that cloud retrievals could be sensitive to veff near the rainbow.

[Figure]

Figure S1: Dependence of GOES-RSP $r_e$ differences relative to RSP effective variance. Absolute differences (left panel), and differences relative to RSP $r_e$.

- Page 7, lines 230-243: Perhaps the MODIS vs GOES re retrieval differences are tied to the rather large central wavelength difference (3.75 vs 3.9μm) and may point to a different

forward model issue? Specifically, the liquid index of refraction assumed in the calculation of the cloud single scattering properties – see Platnick et al (2020) for a discussion of re sensitivities to refractive index and temperature (Platnick S, Meyer K, Amarasinghe N, Wind G, Hubanks PA, Holz RE. Sensitivity of Multispectral Imager Liquid Water Cloud Microphysical Retrievals to the Index of Refraction. Remote Sensing. 2020; 12(24):4165. https://doi.org/10.3390/rs12244165). Note that paper shows MODIS 3.75µm vs VIIRS 3.7µm re differences on the order of those shown here, though I admit the impacts of heterogeneity are difficult to disentangle. Can the authors at least comment on the implications of this on their MODIS vs GOES results?

We use the liquid index of refraction from Hale and Querry (1973) for water at 25˚C. Regarding differences in the forward model between MODIS and GOES-13, rather than deriving lookup tables based on GOES-13 central wavelengths, we compute lookup tables using weighted-average optical properties based on the spectral response function. This method should minimize the channel differences between GOES and MODIS.

We have added the following paragraph to address the reviewer's comment:

" In addition to pixel resolution and viewing geometry differences, the dissimilar spectral response between MODIS and GOES-13 imager could yield retrieval discrepancies if the sensor differences are not properly accounted for in the algorithm, especially considering the spectrally wider GOES-13 channel. To circumvent this problem, rather than deriving optical properties for the central wavelength, we derive solar reflectances (lookup tables) for GOES-13 using weighted-average optical properties based on the instrument's spectral response function. An aspect more difficult to address is the retrieval dependence on the index of refraction dataset. Platnick et al. (2021) found that retrieval differences that arise from the choice of refractive index dataset could explain $r_e$ differences between MODIS and the Visible Infrared Imaging Radiometer suite (VIIRS) on Suomi NPP over ocean of about 1 µm for the 3.7-µm band. While the use of a specific refractive index dataset needs to be scrutinized, we note that pixel resolution and viewing zenith angle (Figs. 9 and 11a) well could explain most of the 2 µm bias of $r_e$ GOES-13 relative to MODIS."

- Page 8, lines 255-258: This may be more challenging, but I think you can at least plot the MODIS scattering angle distributions within each GOES scattering angle bin (perhaps as an accompanying box plot). That should indicate scattering angle sampling differences. You should also plot Terra and Aqua MODIS separately, since the scattering angle sampling may be quite different.

We appreciate the reviewer's suggestion. We followed the reviewer's recommendation and separated Terra from Aqua in Figure 10.

[Figure]

Figure 10: $r_e$ differences between GOES-13 and Terra MODIS (red) and Aqua MODIS (black) binned in deciles of GOES-13 scattering angle ( $\Theta$). Error bars represent the root mean square difference for each bin.

We also analyzed MODIS scattering angle, but did not include this analysis in the paper because it is not possible to disentangle the scattering angle effect from the viewing zenith angle (Figure S2). On average, MODIS scattering angles are typically less than 140°, with a small fraction of samples with angles >140° and a wide range of viewing zenith angle. To reflect this, we wrote in Section 3.3 the following explanation: "A similar analysis applied to MODIS $\Theta$ is more challenging because the range of MODIS $\Theta$ variability is narrower than GOES, and VZA and $\Theta$ cannot be fully disentangled."

[Figure]

Figure S2: MODIS viewing zenith angle and scattering angle for Terra (blue) and Aqua (red) binned as a function of GOES-16 satellite scattering angle depicted in Figure 12.

- Page 8, line 264-267: While the precipitation likely isn't aliasing into the satellite retrievals, how do the DSDs observed by CDP itself change between precipitating and non-precipitating clouds? If it's significant, it's possible that there may be a correlation with re differences given the assumed veff may deviate more/less from reality.

We compared the CDP effective variance against the precipitation liquid water path from the CIP probe (Figure S3). Interestingly, the amount of near cloud-top precipitation and effective variance are uncorrelated.

[Figure]

Figure S3: Relationship between CDP (cloud mode) effective variance and liquid water path derived from the CIP probe.

- Page 9, line 278-279: Besides spatial resolution differences between GOES-13 and 16, what about scattering angle differences? This was pointed to as a key player in the MODIS vs GOES-13 differences, and GOES-13 and 16 weren't viewing from the same orbital location.

Scattering angles differences between GOES-13 and GOES-16 are modest, less than 0.5° (viewing geometries are nearly identical).

- Page 10, lines 323-325: I don't think investigating veff impacts needs to wait for future work, nor does it require using veff as an additional input to the satellite retrievals (i.e., using various veff in the forward models). As I suggested above, you can simply look at re retrieval differences as a function of RSP veff (or CDP veff). You already have these data from RSP, and can calculate veff quite easily from CDP, so the hypothesis at least can be partially tested here. I suggest the authors do this analysis.

See our previous response to the veff comment.

---

## Referee Report (RR1)

Review of "Evaluation of satellite retrievals of liquid clouds from the GOES-13 Imager and MODIS over the midlatitude North Atlantic during NAAMES campaign", by Painemal et al.

Revised manuscript

I have 2 minor comments before the revised manuscript can be accepted for publication:

1) Lines 142-145

The text reads:

Lastly, we note that the algorithm for deriving satellite $r_e$ differs from the RSP algorithm, in that satellite-based $r_e$ relies on the dependence of shortwave-infrared unpolarized reflectance on $r_e$ (and an assumed value for effective variance, **with reflectance monotonically decreasing with $r_e$**), whereas RSP is based on the dependence of the polarized reflectance on the scattering angle, $r_e$, and effective variance near the rainbow."

The phrase highlighted in bold suggests that reflectance monotonically decreases *as $r_e$ decreases*. Do the authors confirm that they meant the opposite, i.e. "*as $r_e$ increases*"? Perhaps also specify "mid-infrared" reflectance?

2) Line 270: => should be "(Figure 10, red)".

---

## Author Response (AR2)

**Referee #1**

We appreciate the reviewer's final comments. His/her comments are addressed below in red.

The text reads:

Lastly, we note that the algorithm for deriving satellite $r_e$ differs from the RSP algorithm, in that satellite-based $r_e$ relies on the dependence of shortwave-infrared unpolarized reflectance on $r_e$ (and an assumed value for effective variance, **with reflectance monotonically decreasing with $r_e$**), whereas RSP is based on the dependence of the polarized reflectance on the scattering angle, $r_e$, and effective variance near the rainbow."

The phrase highlighted in bold suggests that reflectance monotonically decreases *as $r_e$ decreases*. Do the authors confirm that they meant the opposite, i.e. "*as $r_e$ increases*"? Perhaps also specify "mid-infrared" reflectance?

Response: The sentence is correct, near-infrared reflectivity decreases with $r_e$. We now specify that we are referring to near-infrared reflectivity instead of visible reflectivity.

2) Line 270: => should be "(Figure 10, red)".

Response: The sentence was modified accordingly, thanks.

**Referee #3**

We appreciate the reviewer's additional comments. Our responses are highlighted below in red.

I wish that the authors had commented more on the known differences between polarimetric and bi-spectral retrievals. While is is true that the two retrievals have different vertical weighting sensitivities to the cloud vertical profile this is not the only source of difference when comparing them to one another - especially due to differing sources and causes of bias for each retrieval technique which vary significantly with pixel size [Miller et al. 2018]. At high spatial resolution the two retrievals behave quite similarly, even in spite of their vertical weighting differences (refer to figure 6 of the previous paper).

In addition to the collocated analyses shown in this work, it is also useful to look at how retrievals using both of these techniques behave when made from the same platform and at high spatial resolution - because those are the two biggest sources of bias in the comparisons presented here. It is worth exploring because the pixel growth from RSP to MODIS to GOES-13 introduces a significant source of bias for bispectral CER retrievals. However, RSP produces multiple retrieval products - one based on a nadir-viewing bi-spectral retrieval similar in heritage to the Nakajima & King heritage retrievals, and several other polarimetric based on algorithms described in Alexandrov et al. 2012a,b (cited in this study). A comparison of the two RSP products for low-level liquid water clouds during the ORACLES field campaign ( Figure 2 of Miller et al. 2021) revealed similar statistical biases ranging from 0.5-1 micron higher for the bi-spectral CER - similar to what was shown in Miller et al. 2018. The overall bias shown shown here for NAAMES fits into the context of the comparison from ORACLES, so I think it is valuable to mention that results comparing MODIS and RSP fall only slightly larger than the results shown for a RSP-only comparison of retrievals. One could also do this same analysis for NAAMES, but it is perhaps outside of the scope of the authors work presented here.

R: Regarding the modest impact of the vertical structure (weighting function), we added the following sentence:

"The modest impact of the cloud vertical structure in explaining polarimetric and bi-spectral $r_e$ differences is also supported by 1-D theoretical results in Miller et al. (2018) for retrievals derived at the same pixel resolution."

Regarding the ORACLES results reported in Miller et al. 2021, it is difficult to interpret their results (more specifically their Figure 2), because the analysis shows a negative bias in bi-spectral cloud effective radius, that is, bi-spectral $r_e$ is smaller than its polarimetric counterpart. If our interpretation of Miller et al. is correct, then the ORACLES analysis contradicts our analysis and several papers that arrive to the same conclusion, that is, MODIS overestimates $r_e$. Another aspect that makes the analysis in Miller et al. (2021) difficult to link to our study is the fact that at the typical resolution of the RSP footprint (~ 50 m), 3-D radiative transfer effects will be a dominant source of uncertainty in bi-spectral $r_e$, whereas at 1-km pixel resolution of MODIS, 3-D radiative effects are ameliorated (e.g. Zhang et al., 2011). We agree with the reviewer in that more efforts should be devoted to better characterize uncertainties in MODIS retrievals, but such analysis is beyond the scope of our study.

Zhang, Z., Ackerman, A. S., Feingold, G., Platnick, S., Pincus, R., and Xue, H.: Effects of cloud horizontal inhomogeneity and drizzle on remote sensing of cloud droplet effective radius: Case studies based on large-eddy simulations, *J. Geophys. Res.*, 117, D19208, doi:10.1029/2012JD017655, 2012.